# Green Innovation Strategy and Ambidextrous Green Innovation: The Mediating Effects of Green Supply Chain Integration

**Yongbo Sun and Hui Sun ***

Business School, Beijing Technology and Business University, Beijing 100048, China; sunyb@th.btbu.edu.cn
* Correspondence: 1930601079@st.btbu.edu.cn; Tel.: +86-157-3517-9786

**Abstract:** Although the importance of green innovation strategy has been recognized, in the existing literature the relationship between green innovation strategy and corporate competitive advantage, as well as the relationship between green innovation strategy and green innovation are still unclear. Based on the resource dependence theory, this paper proposes a mediation model to promote ambidextrous green innovation. The relationship between green innovation strategy and ambidextrous green innovation is discussed, and the mediating role of green supply chain integration is investigated. Based on the questionnaire data from manufacturing companies in three developed economic zones in China, a structural equation model is established to verify our hypotheses. The empirical results show that the green innovation strategy has a positive impact on both exploitative and exploratory green innovation, and the impact on exploratory green innovation is greater than that on exploitative green innovation. Green supply chain integration plays a partial intermediating role in green innovation strategy and ambidextrous green innovation. The implementation of green innovation requires not only internal cross-department integration, but also integration with external supply chain partners such as suppliers and customers. By emphasizing the importance of green innovation strategy in the context of sustainable development, this research helps provide effective strategic directions and required capacity structures for companies to successfully implement green innovation practices, and reduces the uncertainty of green innovation. This study expands previous studies and enriches existing green innovation research.

**Keywords:** RDT; green innovation strategy; green supply chain integration; ambidextrous green innovation

## 1. Introduction

As environmental issues such as pollution, energy consumption, the greenhouse effect and haze become increasingly prominent in China, they have become important factors hindering social and economic development and enterprise performance [1]. Achieving a "win-win" situation between economic growth and environmental protection has become a problem that enterprise managers must immediately consider. An increasing number of enterprises regard green innovation as an effective means for gaining a competitive advantage. Green innovation, as a new innovation mode, can effectively deal with pollution prevention and control, energy conservation, green technology upgrading and corporate green management [2], and it is conducive to improving the green image of enterprises, promoting enterprises' sustainable development and obtaining economic benefits.

Green innovation has been divided by most previous scholars into green product innovation and green process innovation [3–5]. Some propose that green innovation also includes green management innovation and green marketing innovation [6]. However, the ambidexterity of green innovation has been neglected by scholars. Therefore, according to the ambidexterity theory, this paper divides green innovation into exploitative green innovation and exploratory green innovation [2,7]. Ambidextrous fractal dimension develops a

new perspective on green innovation, which avoids the classification difficulty of previous green innovation dimensions such as green product innovation, process innovation and other internal staggered superposition, which is helpful for extending and expanding ambidexterity theory and green innovation.

To meet the increasing environmental requirements, enterprises must reconfigure their strategic direction and incorporate environmental responsibility into their developmental strategies and business objectives. Green innovation strategy (GIS) is one of the most important environmental strategies, which reduces the environmental impact of enterprises' production management activities through pollution prevention, product management and the use of clean technology [8]. Previous studies on GIS focus on the impact on corporate performance and competitive advantage. Ge et al. (2018) discussed the relationship between GIS and dynamic capability under the condition of environmental uncertainty, and how these factors promote the competitive advantage of enterprises [9]. However, whether GIS can guide enterprises to carry out ambidextrous green innovation activities has not been proved effectively. This paper provides empirical support for the relationship between GIS and ambidextrous green innovation by exploring their influence mechanism. However, in this process, how an enterprise integrates internal and external effective resources to realize green innovation is also the guarantee to ensure the implementation of GIS.

According to the resource dependence theory (RDT), enterprises can obtain the required resources cooperating with supply chain partners to gain a competitive advantage [10]. As an environmental strategy, GIS guides enterprises to reduce the environmental damage caused by their production and operation activities. However, it is difficult for enterprises to protect the environment independently. RDT posits that enterprises that actively implement environmental strategies will more actively cooperate with their supply chain partners to share knowledge and resources required to solve environmental problems [11]. Green supply chain integration (GSCI) can strengthen the bilateral relationship between enterprises and supply chain partners, facilitate information sharing and promote the implement of green-related activities, thereby reducing operating costs, providing a sustainable competitive advantage, and finally bringing economic benefits to enterprises [10,12]. This is consistent with RDT's view that a company's strategy (such as GIS), taking into account its organizational goals and the competitive environment, produces favorable results (such as achievement of green innovation) by establishing a structure that help the firm allocate, coordinate and implement the resources needed for its strategy (such as GSCI). Therefore, this paper takes the GSCI as the mediating factor between GIS and ambidextrous green innovation, which is committed to breaking the "black box" of the relationship between GIS and ambidextrous green innovation, and discusses the internal mechanism between GIS and ambidextrous green innovation.

This paper aims to empirically test the mediating effect of GIS, GSCI and ambidextrous green innovation. According to the ambidexterity theory, we consider two kinds of ambidextrous green innovation, namely exploitative green innovation and exploratory green innovation, and discuss their relationship with GIS, which proves that GIS has a positive impact on exploratory and exploitative green innovation. On the one hand, it expands the related research on green innovation, on the other hand it extends and expands the ambidexterity theory. The empirical results show that green innovation is an effective means for manufacturing companies to gain competitive advantage and sustainable development. On this basis, this research help companies consider corresponding strategic plans that can effectively implement green innovation and reduce innovation risks, which enriches relater research on driving force of green innovation. Furthermore, based on the RDT, it proves that the three dimensions of GSCI, namely green internal integration, green supplier integration and green customer integration play a part of intermediary role between GIS and ambidextrous green innovation. GIS, as an environmental strategy that can effectively implement green innovation, prompts companies to adjust their own resources and capabilities accordingly. With the support of green supply chain integration capabilities, it will

help companies improve production technology, products and services, so as to form their own core competitiveness. This provides a theoretical perspective for understanding the relationship between GIS and ambidextrous green innovation, makes up for the limitations of the "theoretical black box" research between GIS and green innovation, and enriches the relevant research on supply chain integration and green innovation.

## 2. Literature Review and Research Hypothesis

### 2.1. Resource Dependence Theory

According to RDT, organizational production depends on the acquisition and maintenance of key resources, which leads to the inevitable dependence of an organization on the external environment, dependence that will inevitably lead to interdependent organizational behavior and uncertain result [13]. RDT enables scholars to clearly understand the process of an organization adopting various strategies to adjust its own mechanism, select its environment and adapt to the environment. It requires an organization to effectively manage the demands of external resources [14], so as to reduce uncertainty and dependence. A supply chain network is an effective means for organizations to obtain external resources. There are previous studies that use RDT to explain the supply chain. Tan et al. proposed a multi-dimensional framework to consider the adoption of electronic data interchange (EDI) in supplier management and its impact on information and relationship alliances, pointing out that supply chain members attempt to reduce uncertainty and dependence by establishing cooperative relationships with trading partners [15]. Therefore, RDT provides a basic theoretical framework for explaining that inter-firm behavior can help firms gain competitive advantage and achieve their goals.

This study posits that RDT is valuable for explaining how green innovation strategies promote the realization of green innovation through the internal and external integration of supply chain, but the empirical evidence on this relationship is limited at present. In this paper, we attempt to use RDT to explain the supply chain factors that promote successful green innovation by manufacturing firms. According to RDT, inter-organization relationship is resource-dependent relationship, that can be reduced through resource substitution or mutual cooperation. RDT asserts that enterprises cannot arbitrarily choose their preferred path to achieve their results [16]. Instead, an enterprise must rely on other entities in the environment to obtain the resources needed to achieve its goals. Therefore, the enterprise' strategic planning and its interdependence with partners together shape the subsequent organizational results. RDT suggests that enterprises should strengthen internal and external exchanges and cooperation, thus enabling the enterprise to obtain the critical resources required for development in order to reduce risk and uncertainty [17]. RDT also argues that the establishment of cooperative relationships by enterprises (such as GSCI) constitutes a bridge between organization strategy (such as GIS) and the corresponding organizational results (such as ambidextrous green innovation) [16]. Therefore, this study uses RDT to explain the interrelationships between GIS, GSCI and ambidextrous green innovation.

### 2.2. Ambidextrous Green Innovation

Green innovation (GI) is related to products, processes and services that protect the environment. It is a process in which enterprises continuously carry out and implement green activities such as reducing waste, preventing pollution and improving environmental quality, and finally improve environmental and economic performance [7,18,19]. To cope with environmental challenges, enterprises need to carry out exploitative green innovation and exploratory green innovation simultaneously, which not only reduces the negative impact on the environment, but also improves the enterprises' productivity, and enhance their overall image and reputation, to help them improve their performance and gain a competitive advantages [20]. Based on ambidexterity theory, this paper divides green innovation into exploitative and exploratory green innovation [2,7]. Exploitative green innovation refers to the discovery and use of existing environmental knowledge and

experience to improve current green products, processes and services to meet current market and customer needs [2], and continue to extend existing knowledge and technology to improve the utilization of existing resources; Exploratory green innovation refers to the discovery and use of new environmental knowledge and experience to develop new green products, processes and services in order to meet the potential market and customer demand [2]. In this way, it is difficult to be imitated and surpassed in the short term, thus creating a differentiated competitive advantages for enterprises.

Based on relevant literature, previous research on the influencing factors of green innovation focuses on external factors, such as stakeholder pressure [21,22], environmental regulation [23–25], green supplier [26], external knowledge resources [27] and market demand [28]. Few studies emphasize the internal factors that promote green innovation, such as environmental ethics [29,30] and environmental orientation [31]. GIS is one of the most important environmental strategies and an important internal organizational factor that affects the enterprises' production and operation. Previous studies on GIS mostly focus on its impact on enterprise performance and competitive advantage [32], but the mechanism between GIS and green innovation has not been paid much attention from scholars. In addition, previous studies have confirmed the importance of the co-existence of exploitative innovation and exploratory innovation in the enterprise [33], and ambidextrous organizations usually get better performance. However, the ambidextrous green innovation has not yet attracted the attention of scholars, and how the GIS affects the exploitative green innovation and the exploratory green innovation still needs to be studied.

### 2.3. Green Supply Chain Integration

Green supply chain integration (GSCI) refers to the degree to which manufacturers and supply chain partners carry out strategic cooperation and coordinate the management of internal and inter-organizational processes to reduce environmental impact [10,34]. Based on the existing studies on GSCI, scholars generally agree that GSCI is divided into green internal integration (GII), green supplier integration (GSI) and green customer integration (GCI) [34–36]. GII refers to the degree of mutual communication, information resource sharing and coordination among cross-functional departments in the practice of enterprise environmental management [37,38]. GSI refers to the activities of environmental collaboration, information sharing and joint solution of environmental problems with major suppliers that provide resources needed for green practices [37,38]. GCI refers to activities such as environmental cooperation, information sharing and joint solution environmental problems with key customers who provide enterprises with resources needed for green practices [37,38]. As a structure, GSCI helps enterprises to allocate, coordinate and implement the key resources needed for environmental strategy. When an enterprise's environmental strategy is matched with the GSCI mechanism, it will help the enterprise to achieve its strategic objectives and improve its environmental performance [39].

### 2.4. Green Innovation Strategy and Ambidextrous Green Innovation

GIS refers to the strategy by which enterprises adopt green technology or green management to improve their production and operation activities to reduce the negative environmental impact as well as proactively incorporate environmental responsibility into their strategic planning with the goal of achieving coordination between the external environment and organizational conditions [1,9]. Existing studies have found that a strategic orientation promotes enterprise innovation, and enables enterprises to respond quickly to market changes and meet customer needs. Kortmann and Sebastian discussed the influence of strategy implementation on ambidextrous innovation, and proved that the implementation of a strategic orientation promotes exploitative and exploratory innovation behaviors by enterprises [40]. Exploitative green innovation is aimed at meeting the needs of existing markets and customers, which innovation process through existing knowledge, resources and skills mainly brings short-term economic performance to enterprises. Exploratory green innovation is aimed at adapting to the potential market environment, is

related to the long-term strategic performance of the company, and brings a late-mover advantage to the company. Simultaneously conducting exploratory and exploitative green innovations can encourage enterprises to explore new opportunities for green transformation and development while developing existing capabilities and realizing efficient use of resources. Therefore, this study believes that GIS has a positive impact on ambidextrous green innovation.

First of all, GIS encourages the effective use of raw materials to reduce costs and waste [41]. This requires enterprises energy conservation and clean production by repairing existing technologies (improving efficiency) and introducing new technologies (technical progress), such as adopting resource-saving and environmentally-friendly equipment, instruments and technologies to realize resource recycling and reduce pollutant emissions. Implementing and advancing green technology, eliminates the uncertainty caused by green innovation, realizes small continuous innovation in functions or technology to improve existing or develop new products, processes and services, to achieve the aim of meeting environmental requirements, improves corporate environmental performance, promotes the development of green economy, and finally provides economic benefits [1,30,42]. Secondly, when enterprises adopt GIS, it reflects their proactive behavior towards environmental issues in the management of their economic activities. However, enterprises will be affected by disposable resources in the selection of green management behavior [43]. This kind of social responsibility motives enterprises to increase green investment. In particular, exploratory green innovation requires breakthroughs in existing knowledge and skills. There are higher resource, ability, and structural requirements, which hide huge risks and require higher cost input. Therefore, in the process of green management, an enterprise's internal stakeholders will increase the input of effective resources for green products, processes and services, coordinate the required heterogeneous resources and strengthen their environmental willingness, which is conducive to the integration of organizational resources and reduces the risk of process and output on environmental impact. Therefore, the following hypothesis is proposed:

**Hypothesis 1 (H1).** *GIS is positively associated with (a) exploitative green innovation and (b) exploratory green innovation.*

### 2.5. The Meditating Effect of Green Supply Chain Integration

According to RDT, environment-oriented enterprises are more proactive in seeking cooperation with supply chain partners, acquiring their important knowledge and resources, and collaborating to solve environmental problems [11], helping enterprises to achieve excellent business goals and enhance their competitive advantage [10,44]. Studies by scholars have shown that to meet the needs of organizational strategic goals, enterprises must find and match an appropriate supply chain structure [45,46] that supports strategy, depending on its supply chain integration [39]. RDT suggests that the strategic planning of the enterprise and the interdependence between the enterprise and its partners together shape the subsequent organizational results. RDT claims that enterprises are embedded in various interdependent networks and such interdependence can be reduced through supply chain or value chain practices [16]. This indicates that the corporate GIS will alleviate the resource demand for green innovation through collaboration between internal and external supply chain members. In other words, green innovation strategy encourages enterprises to expand their organizational structure and strengthen the integration between internal functional departments and external supply chain partners (such as key suppliers and customers), so that enterprises can obtain key resources that were previously unavailable, and promote ambidextrous green innovation. Therefore, GSCI may play an intermediary role in GIS and ambidextrous green innovation.

GII is a strategic integration through which an enterprise integrates environmental objectives into its own strategy and management system [10], and effectively allocates, coordinates and utilizes internal environmental resources. The enterprises' resource integration

activities depend on the overall enterprise strategy [46]. Enterprises that actively respond to environmental issues are willing to consider environmental factors in their strategic planning, integrate green development into their policies and development goals [47,48], and take various internal green management actions to reduce adverse environmental impact. To obtain a competitive advantage, enterprises must incorporate environmental goals into their internal stakeholders' performance responsibility and compensation systems [48], strengthen their collective consciousness of the environment, and determine the learning direction of the enterprise, which is conductive to establishing common values and cultivating the staff to obtain the attitude of getting new environmental protection skills, encouraging them to find ways to reduce environmental hazards in their daily production and operational processes [10]. In this environment, knowledge exchange and resource sharing among members is conducive to promoting the enthusiasm of relevant departments to participate in green innovation. Secondly, the enterprise promises to promote innovation [1], so a higher level of internal integration can improve the corporate ability to develop and deal with internal resources for green innovation. On the one hand, the enterprises can promote exploitative green innovation through searching and integrating existing environmental knowledge about customers and markets; on the other hand, they identify and integrate new environmental knowledge and propose new ideas and technologies to promote exploratory green innovation.

Due to the uncertainty of green innovation, it is not enough for companies to only conduct green internal integration. Supplier integration and customer integration must provide necessary heterogeneous resources for the implementation of GIS and reduce the risks and complexity of green innovation. RDT posits that organizational goals and dependencies are interrelated, and that more effective organizational goals lead to higher levels of external organizational dependencies [16]. One of the important ways to achieve organizational goals is to establish strong connections and integration with external partners (such as major suppliers and customers), and manage and control uncertainty by acquiring external resources. GIS, a major environmental strategy of enterprise, can lead to an enterprise's increased dependence on external organizations in the process of their green practices. To manage these dependencies and cope with uncertainty and complexity, companies should cooperate with suppliers and customers to acquire knowledge and resources [49,50]. Supply chain partners should share the risks and costs of greening. Some scholars have pointed out that the cost of developing green practices with suppliers is lower than other green practices [51]. Compared with internal enterprises, suppliers may share more costs, which will bring better economic benefits. On the one hand, the enterprise can achieve partial innovation of products and services on the original basis of the enterprise, and promote exploitative green innovation; on the other hand, using the previously untouched knowledge to realize the development of new products and services, and promote exploratory green innovation. Therefore, the following hypothesis is proposed:

**Hypothesis 2 (H2).** *The impact of green innovation strategy on ambidextrous green innovation is mediated by (a) green internal integration, (b) green supplier integration and (c) green customer integration.*

The theoretical model of this study is illustrated in Figure 1.

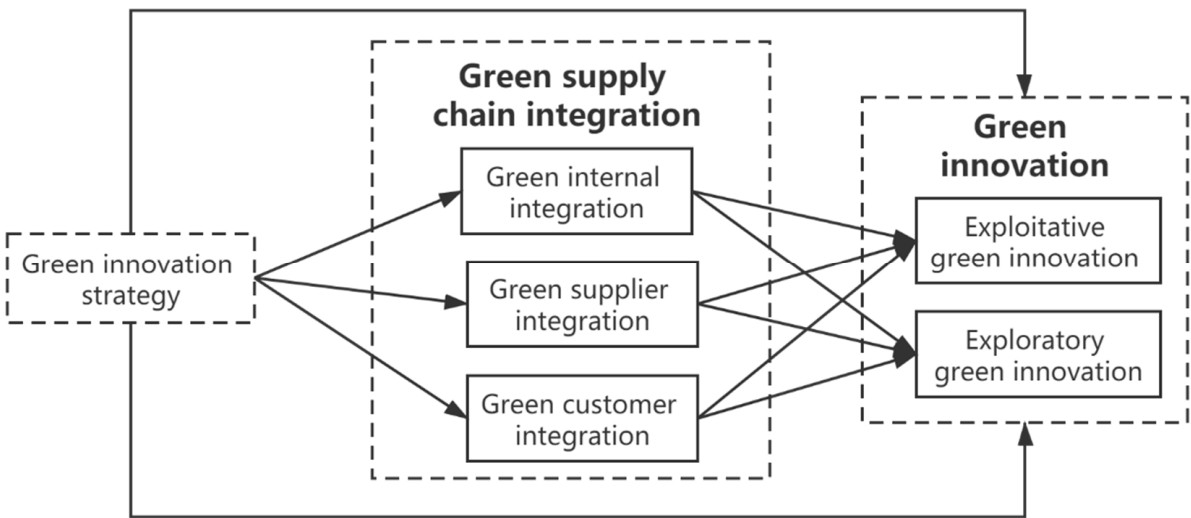

**Figure 1.** The theoretical model.

### 3. Research Methods

*3.1. Sample and Data Collection*

This paper uses a questionnaire survey to collect the data from manufacturing in the three most developed regions of China's economy to verify the hypothesis. In recent years, China has faced serious environmental problems, which the Chinese government attaches great importance to. The regulatory authorities have formulated strict environmental laws and regulations on the production and operation of enterprises, such as the "Law of Environment Protection" and the "Prevention Law of Water Pollution Prevention". In addition, China also has many manufacturing industries that export products abroad and also face strict environmental regulations in exporting countries. Under the strong pressure of domestic and foreign environmental system, Chinese manufacturing enterprises urgently need to reconfigure their strategic direction and capabilities, optimize the existing supply chain process, provide products and services to meet customer needs, and realize sustainable competitive advantage.

We randomly selected manufacturing industries in three economic regions: the Yangtze River Delta, the Pearl River Delta, and the Circum-Bohai Sea Economic Zone as research samples. These three regions are the fastest economic growth in China and are the main force driving China's development plan. These three economic zones have certain differences in geographic location, degree of opening to the outside world, and industrial structure. Therefore, the manufacturing enterprises in these three regions represent different degrees of green innovation efforts. They have certain typical characteristics and can provide a reference for the future development of Chinese manufacturing companies. On the basis of the published industry catalog, we randomly selected manufacturing enterprises in the three regions. We contacted the selected companies by telephone to seek their cooperation, explained the purpose of our survey and ensured the confidentiality of the questionnaire, and at the same time determined whether the company is engaged in green innovation. After establishing their intent to cooperate, we issued 215 questionnaires through electronic questionnaires. To improve the accuracy of the data, we once again stated in the questionnaire that companies which actually implement green innovation activities to fill out this questionnaire and required those who filled out the questionnaires to have a clear understanding of their companies' green innovation and GSCI, we targeted employees such as supply chain managers, general managers and other middle and senior managers. After excluding invalid questionnaires with serious information missing, inconsistent answers and regular answers, 166 questionnaires were collected with a recovery rate of 77.2%, of which 70% were middle and senior managers and 30% were grassroots

managers. The specific conditions of the sampled enterprises in this study are presented in Table 1.

**Table 1.** Characteristics of sample companies.

| Sample Characteristics | Category | Number | Percent |
|---|---|---|---|
| Industries | Food and beverage | 19 | 11.4% |
| | Textile and apparel | 21 | 12.7% |
| | Chemical and related products | 12 | 7.2% |
| | Pharmaceutical and medical | 7 | 4.2% |
| | Rubber and plastics | 8 | 4.8% |
| | Nonmetallic mineral products | 3 | 1.8% |
| | Smelting and pressing | 1 | 0.6% |
| | Metal products | 12 | 7.2% |
| | Machinery and engineering | 48 | 28.9% |
| | Electrical machinery and equipment | 29 | 17.5% |
| | Instruments and related products | 3 | 1.8% |
| | Others | 3 | 1.8% |
| Firm size | Less than 100 employees | 12 | 7.2% |
| | 100–499 employees | 65 | 39.2% |
| | 500–999 employees | 47 | 28.3% |
| | 1000–2000 employees | 21 | 12.7% |
| | More than 2000 employees | 21 | 12.7% |
| Firm age | No more than 5 years | 6 | 3.6% |
| | 6–9 years | 26 | 15.7% |
| | 10–20 years | 94 | 56.6% |
| | More than 20 years | 40 | 24.1% |
| Ownership structure | State-owned and collective enterprises | 26 | 15.7% |
| | Private enterprises | 102 | 61.4% |
| | Foreign-invested enterprises | 17 | 10.2% |
| | Joint venture enterprises | 21 | 12.7% |
| Location | Yangtze River Delta | 67 | 40.4% |
| | Pearl River Delta | 43 | 25.9% |
| | Circum-Bohai Sea Economic Zone | 56 | 33.7% |

*3.2. Variable Measurement*

The scales used in this study are all based on existing literature. The English scale was translated into Chinese using the "back translator" program. First, the members of the study group conducted two-way translation of the English scale, and on this basis, the sentences that were significantly different from the original scale were corrected. Then experts in the supply chain field and middle and senior managers of enterprises were invited to evaluate and test the scale items. Finally, the items of the scale were modified and improved according to the feedback. All of the scales used in this paper are Likert-type 7-point measures, with 1 being completely inconsistent and 7 being completely consistent. Table 2 shows all of the items of the scale.

**Table 2.** Reliability and validity.

| Variables | Items | Loading | Cronbach's $\alpha$ | CR | AVE |
|---|---|---|---|---|---|
| GIS (GIS) | GIS1: Your company has adjusted its business activities to reduce the damage to the ecological environment | 0.763 | 0.713 | 0.911 | 0.595 |
| | GIS2: Although not required by government regulations, your company still have taken the initiative to carry out environmental restoration activities | 0.814 | | | |
| | GIS3: Your company has adjusted its business activities to recycle non-renewable resources, chemicals and components | 0.825 | | | |
| | GIS4: Your company have adjusted its business activities to reduce waste of resources and emissions of pollutants | 0.794 | | | |
| | GIS5: Your company has adopted some new energy sources with less pollution instead of traditional fuels | 0.732 | | | |
| | GIS6: Your company have adjusted its business activities to reduce energy consumption | 0.697 | | | |
| | GIS7: Your company have adjusted its business activities to reduce the environmental impact of their products | 0.764 | | | |
| Green internal integration (GII) | GII1: All functions within your company achieve environmental goals collectively | 0.730 | 0.773 | 0.852 | 0.535 |
| | GII2: All functions within your company develop a mutual understanding of responsibilities regarding environmental performance | 0.678 | | | |
| | GII3: All functions within your company work together to reduce environmental impact of our activities | 0.784 | | | |
| | GII4: All functions within your company conduct joint planning to anticipate and resolve environmental-related problems | 0.749 | | | |
| | GII5: All functions within your company make joint decisions about ways to reduce the environmental impact of our products | 0.712 | | | |
| Green supplier integration (GSI) | GSI1: Your company and suppliers achieve environmental goals collectively | 0.685 | 0.832 | 0.834 | 0.502 |
| | GSI2: Your company and suppliers develop a mutual understanding of responsibilities regarding environmental performance | 0.717 | | | |
| | GSI3: Your company and suppliers work together to reduce environmental impact of our activities | 0.658 | | | |
| | GSI4: Your company and suppliers conduct joint planning to anticipate and resolve environmental-related problems | 0.741 | | | |
| | GSI5: Your company and suppliers make joint decisions about ways to reduce the environmental impact of our products | 0.738 | | | |

**Table 2.** *Cont.*

| Variables | Items | Loading | Cronbach's α | CR | AVE |
|---|---|---|---|---|---|
| Green customer integration (GCI) | GCI1: Your company and customers achieve environmental goals collectively | 0.745 | 0.860 | 0.863 | 0.557 |
| | GCI2: Your company and customers develop a mutual understanding of responsibilities regarding environmental performance | 0.732 | | | |
| | GCIS: Your company and customers work together to reduce environmental impact of our activities | 0.720 | | | |
| | GCI4: Your company and customers conduct joint planning to anticipate and resolve environmental-related problems | 0.759 | | | |
| | GCI5: Your company and customers make joint decisions about ways to reduce the environmental impact of our products | 0.774 | | | |
| Exploitative green innovation (EIGI) | EIGI1: Your company actively improves current green products, processes and services | 0.875 | 0.768 | 0.884 | 0.658 |
| | EIGI2: Your company actively adjusts current green products, processes and services | 0.748 | | | |
| | EIGI3: Your company actively strengthens current green market | 0.846 | | | |
| | EIGI4: Your company actively strengthens current green technology | 0.768 | | | |
| Exploratory green innovation (ERGI) | ERGI1: Your company actively adopts new green products, processes and services | 0.732 | 0.778 | 0.866 | 0.619 |
| | ERGI2: Your company actively exploits new green products, processes and services | 0.869 | | | |
| | ERGI3: Your company actively discovers new green market | 0.806 | | | |
| | ERGI4: Your company actively enters new green technology | 0.732 | | | |

(1) Measurement of the independent variable. We adapted the research of Chan [52] to measure GIS, and the scale has seven items. For example, "Your company adjusted its business activities to reduce the damage to the ecological environment" and others.

(2) Measurement of the mediating variables. We adapted the research of Vachon and Klassen [53] to measure the GII, GSI and GCI, and the scale has four items for each variable. GII includes, for example, "All functions within your company achieve environmental goals collectively" and others; GSI includes, for example, "Your company and suppliers achieve environmental goals collectively" and others; GCI includes, for example, "Your company and customers achieve environmental goals collectively" and others.

(3) Measurements of the dependent variables. We adapted the research of He and Wang [54] and Jansen and others [55] to measure exploitative green innovation and exploratory green innovation, and the scale has four items for each variable. Exploitable green innovation includes such things as "Your company actively improves current green products, processes and services" and others; Exploratory green innovation includes such things as "Your company actively adopts new green products, processes and services" and others.

(4) Control variables. According to relevant previous studies on green supply chain and green innovation, it has been determined that firm size and age will affect enterprise



resource investment and environmental protection activities. Therefore, we use firm size and age as control variables.

## 4. Data Analysis and Results

### 4.1. Reliability and Validity

In this paper, SPSS 22.0 analysis software was used to test the reliability of the scale, and the reliability of the scale was evaluated by examining the internal consistency coefficient (Cronbach's $\alpha$), combined reliability (CR) and average variance extracted (AVE). Factor load, Cronbach's $\alpha$, CR and AVE of each variable are shown in Table 2. As can be seen from the table, the factor load of each item is greater than 0.5, Cronbach's $\alpha$ of each scale is greater than the critical value of 0.7, and CR is greater than the standard value of 0.6, indicating a high degree of consistency of potential variables. AVE is greater than the recommended 0.5, indicating that the scale used in this study has good convergence validity and high reliability.

### 4.2. Confirmatory Factor Analysis

This paper uses the AMOS 20 software to conduct confirmatory factor analysis on GIS, GII, GSI, GCI, EIGI and ERGI to test the discriminative validity of the variables. The results as shown in Table 3, the six-factor model fit of the data ($\chi^2$ =593.621, $\chi^2/df$ =1.534, *RMSEA* = 0.057, *CFI* = 0.902, IFI = 0.904) is significantly better than other nested models. In addition, as shown in Table 4, the square root of AVE of the six variables in this paper is greater than the absolute value of the correlation coefficient between each latent variable, which indicates that the variables have good discriminative validity.

**Table 3.** Confirmatory factor analysis.

| Model | $\chi^2$ | *df* | $\chi^2/df$ | *RMSEA* | *CFI* | **IFI** |
|---|---|---|---|---|---|---|
| Six-factor model | 593.621 | 387 | 1.534 | 0.057 | 0.902 | 0.904 |
| Five-factor model | 684.705 | 392 | 1.747 | 0.067 | 0.860 | 0.863 |
| Four-factor model | 691.548 | 396 | 1.746 | 0.067 | 0.859 | 0.862 |
| Three-factor model | 764.821 | 399 | 1.917 | 0.075 | 0.826 | 0.829 |
| Two-factor model | 797.822 | 401 | 1.990 | 0.077 | 0.811 | 0.814 |
| One-factor model | 852.703 | 402 | 2.121 | 0.082 | 0.785 | 0.788 |

*Notes*: Six-factor model: GIS, GII, GSI, GCI, EIGI, ERGI; Five-factor model: GIS, GII, GSI+ GCI, EIGI, ERGI; Four-factor model: GIS, GII, GSI+GCI, EIGRGI; Three-factor model: GIS, GII+GSI+GCI, EIGI+ERGI; Two-factor model: GIS+GII+GSI+GCI, EIGI+ERGI; One-factor model: GIS+GII+GSI+GCI+EIGI+ERGI.

**Table 4.** Descriptive statistical analysis results.

| Variable | 1 | 2 | 3 | 4 | 5 | 6 | 7 | 8 |
|---|---|---|---|---|---|---|---|---|
| 1.Firm scale | - | | | | | | | |
| 2.Firm age | 0.485 ** | - | | | | | | |
| 3.GIS | 0.197 * | 0.030 | **0.771** | | | | | |
| 4.GII | 0.110 | 0.024 | 0.7 ** | **0.731** | | | | |
| 5.GSI | 0.117 | 0.096 | 0.646 ** | 0.685 ** | **0.709** | | | |
| 6.GCI | 0.028 | 0.060 | 0.479 ** | 0.590 ** | 0.649 ** | **0.746** | | |
| 7.EIGI | 0.155 * | −0.019 | 0.681 ** | 0.688 ** | 0.565 ** | 0.455 ** | **0.811** | |
| 8.ERGI | 0.195 * | −0.013 | 0.678 ** | 0.664 ** | 0.592 ** | 0.490 ** | 0.774 ** | **0.787** |
| Mean | 2.84 | 3.01 | 5.45 | 5.49 | 5.22 | 5.21 | 5.60 | 5.50 |
| SD | 1.139 | 0.738 | 0.688 | 0.831 | 0.989 | 1.045 | 0.835 | 0.937 |

Note: Diagonal entries (in bold) are the square root of the AVE, entries below the diagonal are correlations. ** $p < 0.01$, * $p < 0.05$.

### 4.3. Descriptive Statistical Analysis

With the help of SPSS 22.0, this paper conducted a descriptive statistical analysis of each variable, and the mean value, standard deviation and correlation coefficient of the variables are shown in Table 4. As can be seen from Table 4, GIS is significantly positively correlated with EIGI (r = 0.681, *p* < 0.01) and ERGI (r = 0.678, *p* < 0.01). GIS is significantly positively correlated with GII (r = 0.7, *p* < 0.01), GSI (r = 0.646, *p* < 0.01) and GCI (r = 0.479, *p* < 0.01). GII (r = 0.688, *p* < 0.01), GSI (r = 0.565, *p* < 0.01) and GCI (r = 0.455, *p* < 0.01) are significantly positively correlated with EIGI. GII (r = 0.664, *p* < 0.01), GSI (r = 0.592, *p* < 0.01) and GCI (r = 0.490, *p* < 0.01) are significantly positively correlated with ERGI. The above statistical analysis results provide preliminary support for further demonstration of the hypothesis in this study.

### 4.4. Common Method Variance Test

Since all variables are filled out by the same interviewee, there may be a common method variance (CMV) problem. To reduce the possibility of such variance, this paper uses two statistical verification methods. First, a Harman single-factor test, which is generally accepted by most studies, was conducted whereby all of the items in the questionnaire were subjected to unrotated factor analysis. The variance contribution rate with the highest degree of explanation is 36.5%, which is less than the recommended 40%. This suggests that no single variable explains most of the mutations, therefore, the common method variance is not serious. Second, the results of confirmatory factor analysis show that the indicators of the single-factor model were poorly fitted and did not reach the recommended level, indicating that the deviation of the common method in this paper was not significant.

### 4.5. Hypotheses Testing

To verify the hypothetical model in Figure 1, this paper uses the method of structural equation modeling and the analysis software AMOS 20.0 to compare the theoretical model, the nested model and the alternative model, to find the optimal model. Table 5 shows the test results of them. The nested model deletes the path from GIS to EIGI and ERGI in the theoretical model. There is no mediating effect in the alternative model. GIS, GII, GSI and GCI all have direct influence on EIGI and ERGI. First, we compare the theoretical model with the nested model. From the fitting index, the fit of the theoretical model ($\chi^2$ = 599.659, df = 392, $\chi^2$/df = 1.530, RMSEA = 0.057, CFI = 0.901, IFI = 0.903) and the nested model ($\chi^2$ = 606.354, df = 394, $\chi^2$/df = 1.539, RMSEA = 0.057, CFI = 0.899, IFI = 0.901) are both better. In this paper, we draw on the research method of Anderson et al. (1988) to compare whether the chi-squared change from the theoretical model to the nested model is significant [56]. The results show that the fitting index of the theoretical model is better than that of the nested model. Second, we compare the theoretical model with the alternative model. From the point of view of the fitting index, the fitting index of the alternative model is poor and fails to meet the fitting standard. In conclusion, the theoretical model better reflects the path relationship between the data.

**Table 5.** The fitting index results of the theoretical model, the nested model, the alternative model.

| Model | $\chi^2$ | df | $\chi^2$/df | RMSEA | CFI | IFI |
|---|---|---|---|---|---|---|
| Theoretical model | 599.659 | 392 | 1.530 | 0.057 | 0.901 | 0.903 |
| Nested model | 606.354 | 394 | 1.539 | 0.057 | 0.899 | 0.901 |
| Alternative model | 941.005 | 395 | 2.382 | 0.092 | 0.740 | 0.745 |

In this paper, with the help of the SPSS 22.0 analysis software, the main effect and mediating effect of the model were tested by hierarchical regression method. Table 6 shows the test results of the main effect and mediating effect. The results show that after fixing the influence of the control variable, independent variable GIS (β = 0.815, *p* < 0.01) has a significantly positive impact on EIGI, and Model 2 can additionally explain up to 46.7% of

EIGI ($\Delta R^2$ = 0.467); GIS ($\beta$ = 0.897, *p* < 0.01) has a significantly positive impact on ERGI, and the Model 7 can additionally explains up to ERGI ($\Delta R^2$ = 0.415). Therefore, H1a and H1b are supported.

To test the mediating effect, this study performed the three steps of the mediating effect test proposed by Baron et al. (1986) [47]. The first step is to do the regression of the dependent variables to the independent variable, that is, the regression analysis of GIEI and GREI to GIS. Model 2 and Model 7 in Table 6 examine that GIS has a significant impact on EIGI ($\beta$ = 0.815, *p* < 0.01) and ERGI ($\beta$ = 0.897, *p* < 0.01) on the basis of controlling of the size and age of the company respectively, and the impact of GIS on ERGI is greater than EIGI.

The second step is to perform the regression of the intermediary variables to the independent variable, namely the regression analysis of GII, GSI and GCI on GIS. Model 11, Model 12 and Model 13 in Table 6 examine the impact of GIS on GII, GSI and GCI on the basis of controlling of the size and age of the company respectively. The results show that GIS has a significant impact on GII ($\beta$ = 0.853, *p* < 0.01), GSI ($\beta$ = 0.942, *p* < 0.01) and GCI ($\beta$ = 0.759, *p* < 0.01).

The third step is to do the regression analysis of the dependent variables to the intermediary variables and independent variable, namely the regression analysis of EIGI and ERGI respectively on GIS, GII, GSI and GCI. Model 3, Model 4 and Model 5 are respectively based on Model 2 with GII, GSI and GCI to verify the impact on EIGI. The results show that when GIS is include, GII ($\beta$= 0.421, *p* < 0.01), GSI ($\beta$ = 0.193, *p* < 0.01) and GCI ($\beta$ = 0.143, *p* < 0.01) all have a significant impact on EIGI, and GIS has a significant impact on EIGI, with the coefficients are all significantly reduced. Similarly, Model 8, Model 9 and Model 10 are respectively based on Model 7 with GII, GSI and GCI to verify the impact on ERGI. The results show that when GIS is included, GII ($\beta$ = 0.428, *p* < 0.01), GSI ($\beta$ = 0.266, *p* < 0.01) and GCI ($\beta$ = 0.208, *p* < 0.01) all have a significant impact on ERGI, and GIS has a significant impact on ERGI, with the coefficients are all significantly reduced. In conclusion, GII, GSI and GCI play a partial mediating role in GIS and ambidextrous green innovation, supporting H2a, H2b and H2c.

In addition, to improve the statistical effect of the mediating effect, we further adopted the bootstrap mediating effect test program proposed by Preacher et al. (2008) [57], and ran the Process plugin to verify hypothesis H2 with bootstrap re-sampling technology (see Table 7). In this study, sample size B = 2000 was selected to represent the number of random sampling, and the confidence interval setting of 95% represented the degree of confidence. As can be seen from Table 7, under the 95% confidence interval, the CI interval of the indirect action path of the mediation effect test does not contain zero, indicating that there are indeed significant mediating effects between GII (effect = 0.3594), GSI (effect = 0.1816), GCI (effect = 0.1083) and EIGI, as well as GII (effect = 0.3650), GSI (effect = 0.2509), GCI (effect = 0.1579) and ERGI. To further analyze the types of mediating effects, as the intermediary variables are controlled for GII, GSI and GCI, GIS has a significant impact on EIGI and ERGI. The interval shall not contain zero, so GII, GSI and GCI play an intermediary role between GIS and ambidextrous green innovation. To sum up, it is assumed that H2a, H2b and H2c are strengthened.

**Table 6.** Model hierarchical regression results.

| Variables | EIGI | | | | | | ERGI | | | | GII | GSI | GCI |
|---|---|---|---|---|---|---|---|---|---|---|---|---|---|
| | Model 1 | Model 2 | Model 3 | Model 4 | Model 5 | Model 6 | Model 7 | Model 8 | Model 9 | Model 10 | Model 11 | Model 12 | Model 13 |
| Control variable | | | | | | | | | | | | | |
| Firm size | 0.157 * | 0.040 | 0.052 | 0.050 | 0.055 | 0.217 ** | 0.088 | 0.100 | 0.103 | 0.111 * | −0.029 | −0.056 | −0.110 |
| Firm age | −0.139 | −0.074 | −0.085 | −0.102 | −0.095 | −0.179 | −0.108 | −0.118 | −0.146 | −0.138 | 0.024 | 0.143 | 0.145 |
| Independent variable | | | | | | | | | | | | | |
| GIS | | 0.815 ** | 0.456 ** | 0.633 ** | 0.707 ** | | 0.897 ** | 0.532 ** | 0.646 ** | 0.739 ** | 0.853 ** | 0.942** | 0.759 ** |
| Intermediary variable | | | | | | | | | | | | | |
| GII | | | 0.421 ** | | | | | 0.428 ** | | | | | |
| GSI | | | | 0.193 ** | | | | | 0.266 ** | | | | |
| GCI | | | | | 0.143 ** | | | | | 0.208 ** | | | |
| $R^2$ | 0.036 | 0.467 | 0.557 | 0.497 | 0.491 | 0.053 | 0.469 | 0.542 | 0.514 | 0.509 | 0.491 | 0.427 | 0.242 |
| $\Delta R^2$ | 0.036 | 0.432 ** | 0.089 ** | 0.030 ** | 0.024 ** | 0.053 * | 0.415 ** | 0.073 ** | 0.045 ** | 0.041 ** | 0.478 ** | 0.411 ** | 0.239 ** |

Note: ** $p < 0.01$, * $p < 0.05$.

**Table 7.** Bootstrap analysis of mediating effect.

| Path | Indirect Effect | SE | 95% CI | | Direct Effect | SE | 95% CI | |
|---|---|---|---|---|---|---|---|---|
| | | | LLCI | ULCI | | | LLCI | ULCI |
| GIS-GII-EIGI | 0.3594 ** | 0.0869 | 0.2045 | 0.5455 | 0.4556 | 0.0906 | 0.2766 | 0.6346 |
| GIS-GSI-EIGI | 0.1816 ** | 0.0581 | 0.0849 | 0.3150 | 0.6334 | 0.0908 | 0.4540 | 0.8128 |
| GIS-GCI-EIGI | 0.1083 ** | 0.0417 | 0.0347 | 0.2002 | 0.7067 | 0.0800 | 0.5488 | 0.8646 |
| GIS-GII-ERGI | 0.3650 ** | 0.0999 | 0.1866 | 0.5751 | 0.5321 | 0.1034 | 0.3279 | 0.7363 |
| GIS-GSI-ERGI | 0.2509 ** | 0.0754 | 0.0979 | 0.3968 | 0.6463 | 0.1002 | 0.4483 | 0.8442 |
| GIS-GCI-ERGI | 0.1579 ** | 0.0587 | 0.0607 | 0.2918 | 0.7393 | 0.0881 | 0.5652 | 0.9134 |

Note: ** $p < 0.01$.

## 5. Discussion and Conclusions

### 5.1. Discussion

Based on the RDT, this paper explores the mechanism between GIS, GSCI and ambidextrous green innovation, discusses the direct effect of GIS on ambidextrous green innovation, and considers the mediating effect of GSCI in this relationship. By conducting a questionnaire survey on manufacturing companies in China's three developed economic zones, this paper conducts an empirical study on the models and assumptions constructed, and draws the following conclusions.

First of all, GIS has a significant positive impact on exploitative and exploratory green innovation. This conclusion supports the view of Naidoo and Kortmann et al. that "enterprise strategic orientation promotes enterprise innovation" [40,58]. GIS has a greater effect on exploratory green innovation than exploitative green innovation. One possible reason is that even though China currently attaches great importance to environmental pollution, most Chinese manufacturing companies have limited green practices. In order to quickly occupy a leading position in the market and become a leader in the industry's green standards, companies implement GIS to increase their investment in green resources. As a radical innovation, exploratory green innovation bears greater risks than exploitative green innovation. However, it helps to continuously improve the quality of products and services through the exploration and development of new environmental knowledge, technologies and capabilities. The exploratory green innovation is conductive to establish a good green image of the company, which is difficult to be blindly imitated and surpassed in the short term, to meet the potential market and customer needs, and to form a differentiated competitive advantage of the enterprise. According to the existing literature [2,7,59], Chinese manufacturing companies usually carry out both exploitative green innovation and exploratory green innovation at the same time, but the degree of implementation is different.

At present, most of the research on green innovation focuses on the impact of external pressure on green innovation or the impact of green innovation on corporate performance. Most research on green innovation strategy focuses on the impact on corporate competitive advantages, and only a few literature studies relationships between green innovation strategy and green innovation. Song et al. confirmed that the green innovation strategy positively affects green product innovation and green process innovation through the intermediary effect of green creativity and green organization recognition, but no direct effect is found between the green innovation strategy and green innovation [41]. Soewarno et al. is based on this article, and taking the Indonesian manufacturing industry as an example, it is found that the green innovation strategy is positively affecting green innovation, and the relationship between green organization identification and environmental legitimacy is partially intermediate [60]. Based on the ambidexterity theory, this research divides green innovation into exploitative and exploratory green innovation, confirming that the green innovation strategy positively influences exploitative and exploratory green innovation, and inherits and expands previous studies to a certain extent. It provides certain support for the establishment of green innovation strategies for Chinese manufacturing enterprises and the successful implementation of green innovation activities.

Secondly, GSCI plays a partial mediating role in the relationship between GIS and exploitative and exploratory green innovation, indicating that GIS can not only directly promote the realization of green innovation, but also indirectly promotes the smooth progress of green innovation by carrying out GSCI. The implementation of the GIS requires the integration of internal and external resources, such as internal communication and collaboration between the cross-functional departments, and achievement of strategic cooperation with supply chain partners (key suppliers and customers), which helps enterprises to allocate, coordinate and implement the key resources required for environmental strategies, so that they can smoothly execute green innovation activities and reduce the uncertainty of the ambidextrous green innovation. According to the results of the intermediary effect test, this study shows that the intermediary effect of GII is the best. Without the strategic

role of GII, the commitment and investment of GSI and GCI will be limited [4]. GII as a prerequisite can magnify the ability of suppliers and customers integration dealing with the uncertainty of green innovation. Second, companies cooperate with suppliers on the basis of internal integration, and jointly bear the costs and risks of green innovation. Finally, based on the consideration of corporate needs, integrate customers into corporate green time activities to provide customers with more opportunities to help solve social problems.

As explained above, the research on the impact of GIS on green innovation is mainly through the company's environmental atmosphere as an intermediary, ignoring the joint participation and cooperation of internal and external stakeholders. This research emphasizes the relationship between GIS, GSCI and ambidextrous green innovation of manufacturing enterprises, and confirms the benefits brought by the enterprises' GSCI. Previous studies on the causes and consequences of GSCI, on the one hand, proposed that active environmental strategy and GSCI can improve the environmental performance of the organization [39], but ignored GIS as an effective strategy has a potential influence to GSCI. On the other hand, previous studies on the impact of GSCI on green innovation have focused on distinguishing the impact of different dimensions of GSCI on green product and process innovation, and the conclusions drawn in different situations are inconsistent. This study takes GSCI as a structure, and aims to explore the mediating effects of different dimensions of GSCI in strategy and innovation, and proves that under the guidance of corporate strategy, supply chain members can be integrated into the company's environmental initiatives to provide the key resources and capabilities required for exploitative and exploratory green innovation.

*5.2. Theoretical Contributions*

(1) Based on ambidexterity theory, this paper divides green innovation into exploitative and exploratory green innovation, which provides an effective way to balance short-term interests and long-term strategic goals in green innovation management. Most previous studies divide green innovation into green product innovation and process innovation, and some even include green management innovation and green marketing innovation. In this paper, based on ambidexterity theory, green innovation is divided into exploitative green innovation and exploratory green innovation, which to a certain extent solves the internal cross superposition problem of each dimension in the aforementioned classification of green innovation. For example, the process of green product innovation must also involve in technology, management and marketing innovation, so there are certain difficulties in this division angle. The ambidextrous fractal dimension of green innovation not only makes use of existing resources in mature markets, but also explores new products and services in emerging markets, effectively pursuing and realizing simultaneous but contradictory organizational goals. The combination of green innovation and ambidexterity theory is not only a new perspective of green innovation research, but also an extension of ambidexterity theory, which has theoretical construction and practical guiding significance for the in-depth study of green innovation and its effective implementation.

(2) This paper enriches the research on the antecedents of GIS. Thus far, most relevant studies on green innovation have focused on external pressure and demand, but few studies exist on antecedent variables such as internal strategy. In addition, most of the existing literature on ambidextrous green innovation focuses on the question of whether exploitative and exploratory green innovation can have both, while the common antecedent variables that influence ambidextrous green innovation in the same direction are not given much attention. Therefore, this study deeply explores the influence of GIS on ambidextrous green innovation, verifies that GIS of the enterprise is the key factor in the smooth progress of ambidextrous green innovation, enriches related research on the driving force of green innovation, and has certain theoretical significance for the research of developing GIS to drive green innovation.

(3) Based on RDT, this paper enriches the research on green resource acquisition in the green innovation practices of enterprises. This paper constructs a model that enterprises obtain effective resources to implement GIS (strategy) and promote green innovation activities (results) through GSCI (structure) by integrating RDT and the related green supply chain literature. The empirical results demonstrate that the implementation of green innovation by enterprises requires not only internal integration, but also the integration of external supply chain partners. External integration is helpful for the organization to obtain the complementary resources needed for green innovation, and inter-departmental collaboration within the enterprise is helpful for the implementation of green innovation. GSCI plays an intermediary role in the GIS and ambidextrous green innovation. This result enriches the current research results, and provides a more comprehensive understanding of green supply chain integration, revealing the mechanism of GIS to achieve green innovation under the support of GSCI, and making up for the current lack of research on the "theoretical black box" between GIS and green innovation. As far as we know, there are no studies that examine the GSCI capability in the context of GIS and ambidextrous green innovation. This study fills this gap and makes an important contribution.

*5.3. Managerial Implications*

(1) China's rapid economic growth has been accompanied by serious environmental pollution and resource depletion. At present, many enterprises have realized that to cope with the increasingly serious environmental challenges and strict environmental regulations, they need to incorporate environmental management into their long-term development strategies. Enterprise managers should consider environmental policies. Compared with enterprises that passively implement environmental strategies, enterprises that actively implement GIS will have more competitive advantages, because enterprises that actively implement environmental strategies increase their differentiation advantages by smoothly implementing green innovation activities. Enterprises must change their traditional concepts and regard environmental responsibility and green innovation as a new opportunity to improve existing products and services, improve their environmental efficiency and reduce environmental pollution by means of exploitative green innovation. Through exploratory green innovation, enterprises integrate and utilize new resources to creatively participate in green management and technology, develop new products and services, and meet the environmental protection needs of potential customers.

(2) Enterprises should not only promote the smooth progress of green innovation activities from the strategic level, but also obtain effective resources to implement the strategy through GSCI, to reduce the uncertainty associated with green innovation. This paper has confirmed the importance of GSCI, so companies should work with supply chain partners to solve environmental problems. First, the company must strengthen communication and collaboration between cross-functional departments, reduce contradictions and conflicts between different functions, and ensure the effective flow of resources related to environmental protection among different departments, which lays the foundation for the cooperation between the company and external supply chain partners. In addition, companies should establish a sound environmental management system, environmental protection reward and punishment system, which are conductive to create an environmentally friendly corporate atmosphere, thereby enhancing employees' environmental awareness. Second, companies not only need to select suppliers that meet their environmental standards, but also need to conduct in-depth cooperation with key suppliers to jointly formulate environmental goals, such as joint development of environmentally friendly materials, so as to comprehensively reduce the adverse impact on the environment. Finally, enterprises should obtain the current and future product and service needs of key customers, collect various accurate information related to customers' environmental

needs and preferences through coordination and communication with customers, work together to maintain a good bilateral relationship, and ultimately promote the successful implementation of exploitation and exploratory green innovation activities.

*5.4. Limitations and Future Research*

There are some limitations in this paper. First, the green innovation does not innovate on a single level, but involves all links in the entire life cycle of products, processes, services and management. However, the cross-sectional data adopted in this article ignore the process of dynamic development and changes of the GSCI, exploitative and exploratory green innovation, namely the data of the same variables are collected in a single point in time. Future research should be designed for longitudinal tracking, thereby more accurately revealing the causal relationships between variables and the change process. Secondly, this paper examines the relationship between GIS and ambidextrous green innovation from the perspective of supply chain. Future research should explore other mediating factors such as alliance combination that affect GIS and ambidextrous green innovation. Finally, future research can explore the moderating variables that affect GIS and ambidextrous green innovation.

**Author Contributions:** Conceptualization, Y.S.; Formal analysis, H.S.; Funding acquisition, Y.S.; Investigation, H.S.; Methodology, Y.S.; Writing—original draft, H.S.; Writing—review & editing, Y.S. and H.S. All authors have read and agreed to the published version of the manuscript.

**Funding:** Chinese National Funding of Social Sciences: 18BGL083 Chinese National Funding of Social Sciences: 18BGL083.

**Institutional Review Board Statement:** Not applicable.

**Informed Consent Statement:** Not applicable.

**Data Availability Statement:** Not applicable.

**Conflicts of Interest:** The authors declare no conflict of interest.

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
