# Peer review of "Green Innovation Strategy and Ambidextrous Green Innovation: The Mediating Effects of Green Supply Chain Integration"

_sustainability, doi:10.3390/su13094876_

Round 1
Reviewer 1 Report
Rather long introduction which could be clearer about what the contribution(s) of the paper is.
Some of the hypotheses appear trivial. If a firm has a green innovation strategy, it would be very surprising if it would not carry out some innovation activities in one way or another. So, the positive relationships in H1-H5, are not terribly novel. In fact, what is missing is trade-offs. A firm cannot choose to do whatever it likes; some activities are more productive or more costly than others. If a firm has to choose between internal/supplier/customer innovation integration, which one does it choose? The same applies to exploitative vs explorative innovation. Not all firms need to engage in both; that varies according to competition in the market, which industry we are talking about, etc. The theory is described as if there are no costs, no trade-offs; just “pick what’s best for you”. Very little resemblance to real firms.
The statistical analysis is competently carried out and the authors are no doubt well into the field of SEM. However, that does not help when the data (survey) collected are highly unlikely to be representative of Chinese manufacturing firms in general. First, the authors contacted companies in three of the most advanced economic zones in China, where environmental problems are higher on the agenda, and secondly, the firms are told the purpose of the study and can choose not to participate. (Moreover, a significant portion of questionnaires were not usable.) Thus, this is a highly selected sample with a clear green profile bias. There are not likely to many firms producing using coal or inputs produced in a non-ecological manner, which still constitute the bulk of China’s manufacturing sector.
The statements, respondents are to take on a stand on using a Likert scale, could have been phrased in a more neutral language. The signal is that “green innovation, production,….” is positive; several statements remind of “leading questions” in journalism. Makes one worried that respondents can have been influenced. You get what psychologists call “demand bias”.
A key feature of the analysis is to account for ambidextrous innovation. However, readers are not informed how large a proportion of respondent firms are engaged in both exploitative and explorative innovation activities, nor how many in only one of these two. Consequently, when you find that GIS is positively correlated with both exploitative and explorative innovation activities you cannot necessarily draw the conclusion that GIS is positively associated with ambidextrous green innovation. In fact, the authors find that the impact is larger for exploitative innovation activities, which is what one would expect. But whether GIS increases exploitative and explorative innovation activities in the same firm, we do not know. Hence, the mediation analysis suffers from the same problem.
Managerial implications are about what managers/firms should do. But the study argues that the firms examined already do that. So, whom are you talking to?
Author Response
Dear reviewer:
Thank you for your valuable comments on our manuscript. Those comments are extremely valuable to our paper, and significant to our researches. We have studied comments carefully and have made correction which we hope meet with approval. We use the ‘Track Changes’ function in Microsoft Word and revised portion marked in red in the paper. The following is the specific modification instructions, please review them.
Modification instructions
1.Line 6 and Line 7: Modified email address
2.Line 21-24:Added research significance
3.Line 78-98:Added research significance
4.Line 178-186: Added some sentences
5.Line 198: Added reference 38
6.Line 202-204: Added some sentences
7.Line 204-209:Modified English expression
8.Line 243-253:Added some sentences
9.Line 244: Added references 4 and 49
10.Line 247: Added reference 50
11.Line 250: Added reference 49
12.Line 261-267: Modified English expression
13.Line 260 and Line 263: Added reference 52
14.Line 272-273: Added some sentences
15.Line 283-286: Added some sentences
16.Line 283: Added reference 56
17.Line 285: Added reference 57
18.Line 393-396: Added some sentences
19.Line 570: Discussion part of the structure has changed
20.Line 585-591: Added some sentences
21.Line 592-609: Added a paragraph
22.Line 299: Added reference 36
23.Line 603: Added reference 67
24.Line 613-619: Modified English expression
25.Line 619-628: Added some sentences
- Line 622: Added reference 4
27.Line 629-644: Added a paragraph
28.Line 636: Added reference 45
29.Line 717-728: Modified content
The remaining changes in the submitted modified manuscript of the word are for the polish of the English expression.
Best wishes to you!

Reviewer 2 Report
The paper is thoroughly written and documented, presenting a relevant research on a topic in a continuous development.
First of all, the introduction and theoretical background is well documented, starting at line 39, with information based on green product innovation, as well as on green process innovation.
Additionally, the literature review is consistent enough, the author not confining himself just to summarize some references, but discussing them critically. As a result, the author evaluated both old interpretations such as Pfeffer, J. & Salancik, G.R. (1979), Baron, R. M. & Kenny, D. A. (1986), Chavez et al. (1996), combined with new scientific materials:
Sharma, S. (2000), Eiadat Y. et al. (2008), Chang, C. H. (2011), or current research: Feng, Z. & Chen W. (2018), Shah, S. A. A. et al. (2020), Sharma S. (2020).
I recommend to expand the literature you have used to develop your theoretical and methodological framework. Some more bibliography can be added on the subject:
Kraus, S; Rehman, SU, Garcia, FJS, Corporate social responsibility and environmental performance: The mediating role of environmental strategy and green innovation, TECHNOLOGICAL FORECASTING AND SOCIAL CHANGE, Volume:160, DOI: 10.1016/j.techfore.2020.120262, Published: NOV 2020
Musaad, Almalki Sultan O., Zhuo, Zhang, Musaad, Almalki Otaibi O., Siyal, Zafar Ali, Hashmi, Hammad, Shah, Syed Ahsan Ali (2020), A Fuzzy Multi-Criteria Analysis of Barriers and Policy Strategies for, Small and Medium Enterprises to Adopt Green Innovation, SYMMETRY-BASEL, 12(1), WOS:000516823700116
Khurshid, F, Park, WY, Chan, FTS (2019), Innovation shock, outsourcing strategy, and environmental performance: The roles of prior green innovation experience and knowledge inheritance, BUSINESS STRATEGY AND THE ENVIRONMENT, 28(8), WOS:000516621200007
Nevertheless, in terms of grammar some of the phrases used in the article are very long and not easy to understand. For example:
- Line 179: Therefore, in the process of green management, internal stakeholders of enterprises will increase the input of effective resources for green products, processes and services, coordinate the required heterogeneous resources and strengthen the enterprise's environmental willingness, which is conducive to the integration of organizational resources and reduce the risk of process and output on the environmental impact. Therefore, the following 183 hypotheses are proposed:[...]
- Line 229: In order to obtain competitive advantage, enterprises need to incorporate environmental goals into their internal stakeholders’ performance responsibilities and compensation systems, strengthen their collective consciousness of the environment, and determine the learning direction of the enterprise, which is conductive to establishing common values and cultivating all the staff to obtain the attitude of getting new skills on the environmental protection, encouraging them looking for ways to reduce environmental hazards in their daily production operation process.
- Line 560: The implementation of the green innovation strategy need to integrate internal and external resources, such as internal communication and collaboration between the cross-functional departments, and achieve strategic cooperation with supply chain partners (key suppliers and customers), which help enterprises allocate, coordinate and implement key resources required for environmental strategies, so that enterprises can smoothly carry out green innovation activities and reduce the uncertainty of the ambidextrous green innovation.
Secondly, the methodology used in the paper is explained properly. For instance, lines 342-406 presents in detail the entire process used for research, including here the questionnaire methodology, along with the research hypotheses. Specifically, 215 electronic questionnaires were used in the research, and 166 questionnaires were considered valid for the study.
Equally important, the results (starting from line 408) and the discussion section are clearly presented including all the details necessary for a good understanding of the content.
Finally, the conclusions of the study are tied with the results and the objectives of the research. Specifically, the author included here the conclusions and arguments relating to every result obtained.
I consider that the paper is publishable after minor revisions regarding the enrichment of bibliographic references and the revision of some sentences.
Author Response

(The authors gave the same response as above.)

Reviewer 3 Report
This is an interesting paper focusing on an area where there is need for more knowledge. The paper addresses the area that refers to the issue of creating green innovations by the industrial companies operating in China. The topic is sound and into the scope of the Journal. The paper is written in a clear and fluent way and the analysis conducted appear to be thorough and rigorous. The methodological approach is well described and suitable for the task at hand. The content of the article corresponds to its aim - substantive compatibility in theoretical, methodological and empirical terms. The research aim of the study has been achieved. The results of the analysis are well presented. The identified limitations for the conducted research inquiry were shown.
Some minor comments:
Both Abstract and Introduction lack clear statement of the aim of the research. I suggest to clearly emphasize the aim of the manuscript both in the abstract and the Introduction section. I also invite the authors to emphasize the significance of the conducted study. Thus, in the Introduction, you should say what kind of use you envisage for the study results.
Discussion section has been located as part of Conclusions. Please mark out Discussion section as a separate paper’s chapter. In my opinion the Discussion part of the paper is too short. This section is crucial for academic papers and should be about contrasting the manuscript's findings with the existing literature. Please supplement Discussion section with more information to what extent your results are aligned with the evidence found in the adequate literature. To what extent your results are not aligned? Please provide a more comprehensive discussion on this.
In short, important good research work well presented.
Author Response

(The authors gave the same response as above.)

Round 2
Reviewer 1 Report
It has not improved much at all. The serious concerns I had regarding fundamental problems with the data (and key variables) and the statistical analysis which does not support their conclusions still remain. In fact, they have not been addressed. And seriously, it could not be done in such a short time. Thus, the arguments for the rejection of the submitted paper remain and have actually been strengthened by the resubmission. Consequently, I recommend rejection.Author Response
Dear Reviewer:
I am very grateful to your comments for the manuscript. We are sorry that the manuscript submitted for the first time did not solve your problem well. We apologize for the inconvenience caused. In order to further improve the quality of the article and solve your doubts about this article, according with your advice, we amended the relevant part in manuscript again. Some of your questions were answered below, and we sincerely hope that this time the manuscript can solve your proposed problem to a certain extent.
The following content is specific modification instructions, please review them.
1.Line 14: Corrected the expression error
2.Line17:Corrected the research conclusion
3.Line76-84:Added the internal mechanism of GSCI in GIS and ambidextrous green innovation
4.Line176-192:Changed the previous 2.4.1 to 2.3 Green Supply Chain Integration
5.Line237-301:The previous 2.4 The Meditating Effect of Green Supply Chain Integration was changed to 2.5, and the research hypothesis was deleted and revised.
6.Line307-308:Modified and deleted content expression
7.Line319-326:Modified and supplemented the reason for the selected area
8.Line330-331,332-334:Supplemented the questionnaire requirements
9.Line455:Changed Model 10 to Model 7
10.Line456:Modified b value
11.Line457:Corrected the research conclusion
12.Line464:Removed the previous hypothesis verification
13.Line467-477:Modified the corresponding model numbers and assumptions according to Table 6
14.Line493:Deleted the content of the table
15.Line500,502,505:Corrected the expression error
16.Line507-519:Corrected the research conclusion and corrected the possible reasons for the conclusion.
17.Line519-522:Deleted and added the content
18.Line542:Corrected the expression error
